# A neuropeptide regulates fighting behavior in *Drosophila melanogaster*

**Fengming Wu[1,2†], Bowen Deng[3,4†], Na Xiao[5†], Tao Wang[1,6], Yining Li[3,7], Rencong Wang[1,2], Kai Shi[1,2], Dong-Gen Luo[4,5], Yi Rao[3,4,7], Chuan Zhou[1,2,4*]**

[1]State Key Laboratory of Integrated Management of Pest Insects and Rodents, Institute of Zoology, Chinese Academy of Sciences, Beijing, China; [2]University of Chinese Academy of Sciences, Beijing, China; [3]Chinese Institute for Brain Research, Peking-Tsinghua Center for Life Sciences, Zhongguangchun Life Sciences Park, Beijing, China; [4]Institute of Molecular Physiology, Shenzhen Bay Laboratory, Shenzhen, China; [5]State Key Laboratory of Membrane Biology, College of Life Sciences, IDG/McGovern Institute for Brain Research, Peking-Tsinghua Center for Life Sciences, Academy for Advanced Interdisciplinary Studies, Center for Quantitative Biology, Academy for Advanced Interdisciplinary Studies, Peking University, Beijing, China; [6]School of Life Sciences, University of Science and Technology of China, Hefei, China; [7]Peking-Tsinghua Center for Life Sciences, PKU-IDG/McGovern Institute for Brain Research, Advanced Innovation Center for Genomics, Peking University School of Life Sciences, Beijing, China

**Abstract** Aggressive behavior is regulated by various neuromodulators such as neuropeptides and biogenic amines. Here we found that the neuropeptide *Drosulfakinin (Dsk)* modulates aggression in *Drosophila melanogaster*. Knock-out of *Dsk* or *Dsk* receptor *CCKLR-17D1* reduced aggression. Activation and inactivation of Dsk-expressing neurons increased and decreased male aggressive behavior, respectively. Moreover, data from transsynaptic tracing, electrophysiology and behavioral epistasis reveal that Dsk-expressing neurons function downstream of a subset of P1 neurons (*P1ᵃ-splitGAL4*) to control fighting behavior. In addition, winners show increased calcium activity in Dsk-expressing neurons. Conditional overexpression of Dsk promotes social dominance, suggesting a positive correlation between Dsk signaling and winning effects. The mammalian ortholog CCK has been implicated in mammal aggression, thus our work suggests a conserved neuromodulatory system for the modulation of aggressive behavior.

**\*For correspondence:**
zhouchuan@ioz.ac.cn

[†]These authors contributed equally to this work

**Competing interests:** The authors declare that no competing interests exist.

## Introduction

Aggression is a common innate behavior in most vertebrate and invertebrate species and a major driving force for natural and sexual selections (*Darwin, 1859*; *Lorenz, 1963*). It is a critical behavior for defense against conspecifics to obtain food resources and mating partners (*Beye et al., 1998*; *Kravitz and Huber, 2003*; *Siegel et al., 1997*).

Aggressive behavior of fruit flies was first reported by Alfred Sturtevant (*Sturtevant, 1915*). Since then, a number of ethological and behavioral studies in flies pave the way for using *Drosophila* as a genetic system to study aggression (*Chen et al., 2002*; *Hoffmann, 1987*; *Hoffmann and Cacoyianni, 1989*; *Hoffmann and Cacoyianni, 1990*. *Drosophila* provides an excellent system to manipulate genes and genetically defined populations of neurons, leading to the identification of multiple genes and neural circuits that control aggression. The neural circuits of aggression involve the peripheral sensory systems that detect male-specific pheromones and auditory cues necessary for aggression (*Liu et al., 2011*; *Versteven et al., 2017*; *Wang and Anderson, 2010*; *Wang et al.,*

*2011*), a subset of P1 neurons (*Hoopfer et al., 2015*), pCd (*Jung et al., 2020*) in the central brain controlling aggressive arousal, and AIP neurons controlling threat displays (*Duistermars et al., 2018*). Aggression is modulated by various monoamines and neuropeptides. Octopamine, serotonin and dopamine are important neuromodulators for fly aggression and the specific aminergic neurons that control aggression have been identified (*Alekseyenko et al., 2014*; *Alekseyenko et al., 2013*; *Certel et al., 2007*; *Hoyer et al., 2008*; *Watanabe et al., 2017*; *Zhou et al., 2008*). Neuropeptides such as tachykinin and neuropeptide F are required for normal male aggression (*Asahina et al., 2014*; *Dierick and Greenspan, 2007*). Cholecystokinin (CCK) is a neuropeptide that is linked to a number of psychiatric disorders and involved in various emotional behaviors in humans and other mammals (*Arey et al., 2014*; *Sears et al., 2013*; *Shen et al., 2019*; *Tõru et al., 2010*). Infusion of CCK induces panic attack in humans (*Bradwejn et al., 1990*). Enhanced CCK level is detected in a rat model of social defeat (*Becker et al., 2001*; *Becker et al., 2008*). CCK is implicated to act in the periaqueductal gray to potentiate defensive rage behavior in cats (*Luo et al., 1998*). In addition, CCK is a satiety signal in a number of species. Silencing CCK-like peptide Drosulfakinin could decrease satiety signaling and increase intake of food in flies. (*Nässel and Williams, 2014*; *Williams et al., 2014*). Co-injection of nesfatin-1 and CCK8 decreased food intake in Siberian sturgeon (*Acipenser baerii*) (*Zhang et al., 2018*).

Here we investigate the roles of cholecysokinin-like peptide Drosulfakinin (Dsk) in *Drosophila* aggression. We generated knock-outs and GAL4 knock-ins for *Dsk* and candidate *Dsk* receptors. Loss-of-function in either *Dsk* or *Dsk* receptor *CCKLR-17D1* reduces aggression. Thermogenetic activation of $Dsk^{GAL4}$ neurons promotes aggression, while silencing these neurons suppresses aggression. We performed transsynaptic tracing, electrophysiology and behavioral epistasis experiments to illustrate that Dsk-expressing neurons are functionally connected with a subset of P1 neurons ($P1^a$-*splitGAL4*, 8 ~ 10 pairs of P1 Neurons) and act downstream of a subset of P1 neurons to control fighting behavior. Furthermore, we found that winners show increased calcium activity in Dsk-expressing neurons and that conditional overexpression of Dsk promotes winning effects, implicating an important role of the Dsk system in the establishment of social hierarchy during fly fighting. Previously the mamalian ortholog CCK has been implicated in aggression (*Li et al., 2007*; *Luo et al., 1998*), thus our work suggests a potentially conserved neural pathway for the modulation of aggressive behavior.

## Results

### Reduced aggression in Loss-of-function *Dsk* mutants

We used genome editing by the CRISPR-Cas9 system to target the *Dsk* locus and generate knock-out and knock-in lines (*Deng et al., 2019*). The 5′ UTR and coding region of *Dsk* were replaced by a *3P3-RFP* cassette through homologous recombination to obtain the knock-out line, which we refer to as Δ*Dsk* (*Figure 1A*, *Figure 1—figure supplement 1*). Immunohistochemical analysis confirmed that Dsk immunoreactivity is detected in the brains of wildtype and Δ*Dsk/+* but not detected at all in the brain of Δ*Dsk/*Δ*Dsk* (*Figure 1B*). Interestingly, homozygous Δ*Dsk* male mutants showed reduced frequency of lunge (*Figure 1H*) and wing threat (*Figure 1—figure supplement 2*) and prolonged latency to initiate fighting (*Figure 1I*), while conditional overexpression of Dsk promoted aggressive behavior (*Figure 1—figure supplement 3*). Furthermore, female aggression is also suppressed in homozygous Δ*Dsk* mutants (*Figure 1—figure supplement 4*, *Video 1*), suggesting that Dsk is required for aggressive behavior in both sexes. Note that the Δ*Dsk* mutants do not show defects in courtship behavior or locomotion activity, but show increased food intake (*Figure 1—figure supplement 5*).

We next performed genetic rescue experiments to examine whether expression of Dsk in DSK neurons is sufficient to restore the aggressiveness in *Dsk* mutants. To this end, we replaced the coding region of *Dsk* with a *GAL4* cassette to generate the $\Delta Dsk^{GAL4}$ knock-in line, in which *GAL4* transcripts instead of *Dsk* transcripts are produced from *Dsk* promoter (*Figure 1A*). $\Delta Dsk^{GAL4}$ identifies precisely the DSK neurons in the male brain (*Figure 1C*). $\Delta Dsk^{GAL4}$ is a null mutation for *Dsk*, as *Dsk* immunoreactivity is absent in both $\Delta Dsk^{GAL4}/\Delta Dsk$ hetero-allelic (*Figure 1D*) and $\Delta Dsk^{GAL4}/\Delta Dsk^{GAL4}$ homozygotes (*Figure 1E*). $\Delta Dsk^{GAL4}$ driven *UAS-Dsk* expression was capable of restoring the Dsk immunoreactivity in both $\Delta Dsk^{GAL4}/\Delta Dsk^{GAL4}$ homozygous mutant background (*Figure 1F*) and

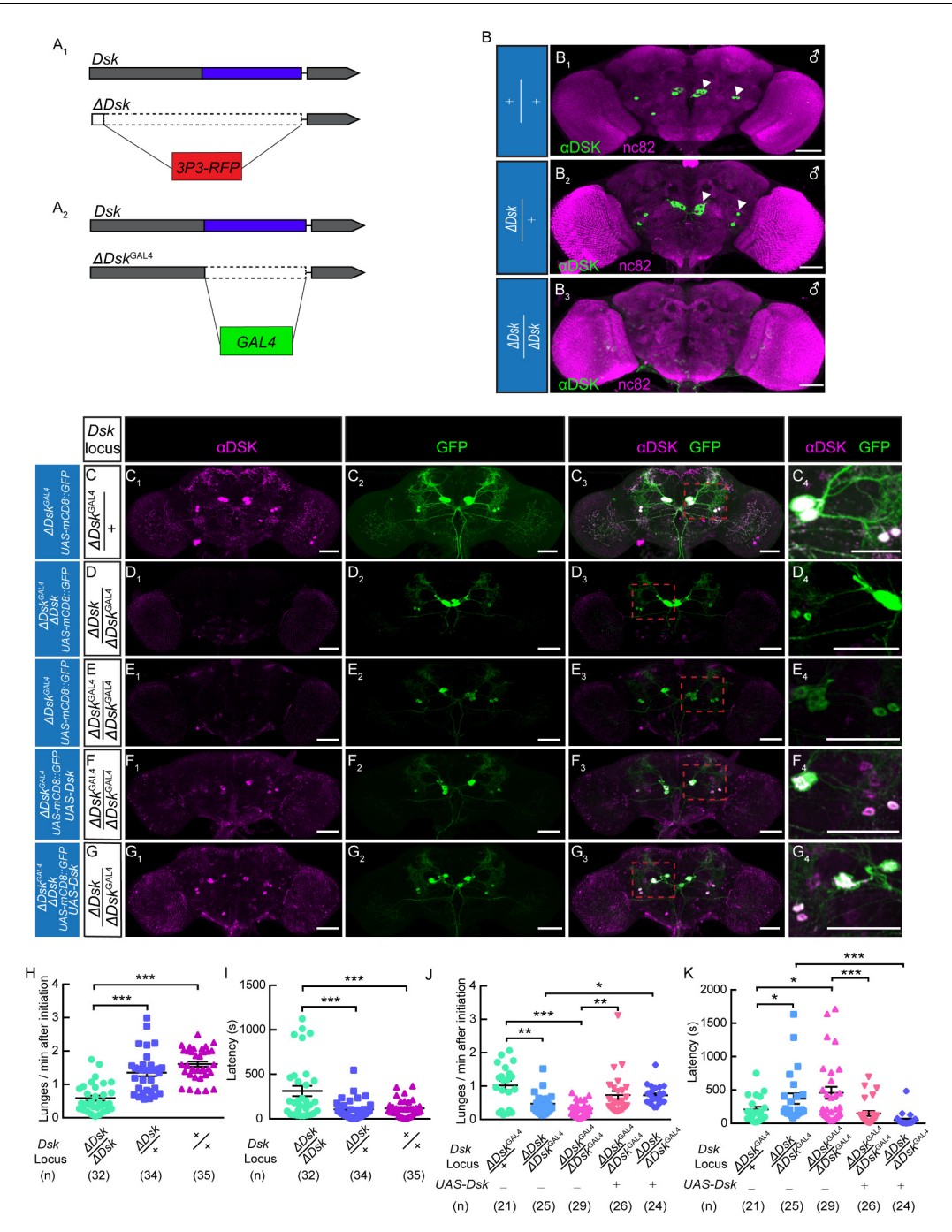

**Figure 1.** The *Dsk* Gene Is Essential for Modulating Male-Male Aggression. (A) Generation of $\Delta Dsk$ (A$_1$) and $\Delta Dsk^{GAL4}$ (A$_2$). Dashed boxes indicate the region replaced by *3P3-RFP* (A$_1$) or *GAL4* (A$_2$) cassette. (B) Male adult brains of the indicated genotypes were stained with anti-DSK antibody (green) and counter-stained with nc82 antibody (magenta) to label neuropil. Arrowheads: Dsk-expressing neurons. (C–G) $\Delta Dsk^{GAL4}$ driven *UAS-mCD8:GFP* expression in *+/$\Delta Dsk^{GAL4}$* (C), $\Delta Dsk/\Delta Dsk^{GAL4}$ (D), $\Delta Dsk^{GAL4}/\Delta Dsk^{GAL4}$ (E), $\Delta Dsk^{GAL4}/\Delta Dsk^{GAL4}$ rescued by *UAS-Dsk* (F), $\Delta Dsk/\Delta Dsk^{GAL4}$ rescued by *UAS-Dsk* (G). Male brains were stained with anti-DSK antibody (magenta; C$_1$–G$_1$) and anti-GFP antibody (green; C$_2$–G$_2$). the anti-DSK antibody signal was undetected in *Dsk* mutant background (D$_1$ and E$_1$), but recovered when the *Dsk* mutant was rescued by $\Delta Dsk^{GAL4}$ driven *UAS-Dsk* expression (F$_1$ and G$_1$). C$_4$-G$_4$: regions in the red dashed boxes of C$_3$-G$_3$. Scale bars represent 50 μm. (H–I) *Dsk* mutants show reduced number of lunges (H) and prolonged fighting latency (I) compared with wildtype and heterozygous controls. (J–K) Number of lunges per minute after initiation (J) and fighting latency (K) for indicated genotype. Reduced aggression phenotypes of *Dsk* mutants were rescued by $\Delta Dsk^{GAL4}$ driven *UAS-Dsk* expression. *p<0.05, **p<0.01, ***p<0.001, n.s. indicates no significant difference (Kruskal-Wallis and post-hoc Mann-Whitney U tests).

The online version of this article includes the following figure supplement(s) for figure 1:

*Figure 1 continued on next page*

**Figure supplement 2.** ΔDsk suppresses wing threat frequency in males.

**Figure supplement 4.** DSK signal modulates female-female aggression.

**Figure supplement 5.** Courtship, locomotion and feeding behavior of Δ*Dsk* mutants.

Δ*Dsk*^GAL4/Δ*Dsk* hetero-allelic mutant background (*Figure 1G*). Next we used these genetic reagents to perform behavioral analysis. As expected, Dsk expression in Δ*Dsk*^GAL4 neurons restores the aggression levels of Δ*Dsk*^GAL4/Δ*Dsk*^GAL4 and Δ*Dsk*^GAL4/Δ*Dsk* mutants to comparable levels observed in heterozygous controls (*Figure 1J and K*).

## Anatomical and behavioral dissection of *Dsk*^GAL4 Neurons

To precisely mark DSK neurons without affecting Dsk expression, a knock-in *Dsk*^GAL4 was generated by fusing the *GAL4* cassette to the end of the *Dsk* open reading frame with a T2A peptide linker (*Donnelly et al., 2001*; *Figure 2A*). Eight neurons, with four neurons in the lateral region (DSK-L) and four neurons in the middle region (DSK-M) were labeled by *Dsk*^GAL4 driving the expression of UAS-mCD8:GFP in the brain, which fully recapitulated the endogenous expression pattern of Dsk in both female and male brains by co-staining experiments (*Figure 2B and C*).

We further analyzed the projection patterns of individual *Dsk*^GAL4 neurons using the MultiColor Flip-Out (MCFO) method (*Figure 2D*; *Nern et al., 2015*). Morphologically, these *Dsk*^GAL4 neurons could be classified into three neuronal types. The type I neurons project to the lobula and ventrolateral neuropils, and extend descending fibers to innervate the Accessory Mesothoracic Neuropil (AMNp) in the VNC (*Figure 2E*, *Figure 2—figure supplement 1*). The type II neurons arborize in the ventrolateral neuropils and superior lateral protocerebrum (*Figure 2F*, *Figure 2—figure supplement 1*). The arborizations of type III neurons project ipsilaterally to the superior lateral protocerebrum (*Figure 2G*, *Figure 2—figure supplement 1*).

To assess the function of *Dsk*^GAL4 neurons in aggression, we silenced *Dsk*^GAL4 neurons by expressing tetanus toxin light chain (TNT) which can completely block chemical synapses (*Keller et al., 2002*; *Sweeney et al., 1995*; *Watkins et al., 1996*), or an inwardly rectifying K^+ channel (Kir2.1) which can hyperpolarize neurons (*Thum et al., 2006*). Compared to control males, male flies with *Dsk*^GAL4 neurons inhibited by either TNT or Kir2.1 showed reduced lunge frequency and prolonged fighting latency (*Figure 2H–K*). Conversely, we activated the *Dsk*^GAL4 neurons by using the temperature sensitive cation channel *Drosophila* TRPA1 (dTRPA1) (*Hamada et al., 2008*). Males carrying *Dsk*^GAL4 and *UAS-dTRPA1* displayed higher aggression level at 28℃ than at 21℃, and the temperature-dependent changes of aggression levels were not observed in males carrying *Dsk*^GAL4 or *UAS-dTRPA1* alone (*Figure 2L–M*; *Video 2*). In addition, no significant changes of locomotion activity were observed after the inactivation or activation of *Dsk*^GAL4 neurons (*Figure 2—figure supplement 2*). Thus, these results support a role of Dsk-expressing neurons in modulating intermale aggression.

To characterize the subset of Dsk-expressing neurons involved in aggression, we generated 93 mosaic flies in which subset of *Dsk*^GAL4 neurons were inactivated by the expression of Kir2.1::eGFP and paired each mosaic fly with a Δ*Dsk* mutant fly to quantify the lunge frequency. We then determined which *Dsk*^GAL4 neurons are labeled in all 93 mosaic flies (*Supplementary file 2*). We found there is a negative correlation between the number of type I plus type II neurons inhibited and the lunge numbers of individual flies (*Figure 2—figure supplement 3A*).The

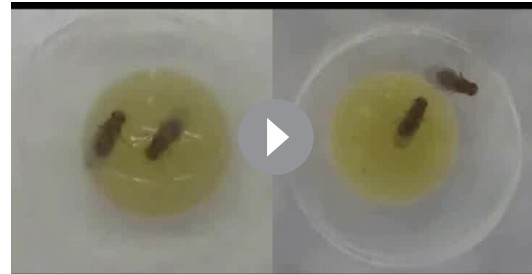

**Video 1.** Left, female aggression of Δ*Dsk* mutants. Right, female aggression of wildtypes. Female Δ*Dsk* mutants show fewer aggressive encounters than wildtypes.

https://elifesciences.org/articles/54229#video1

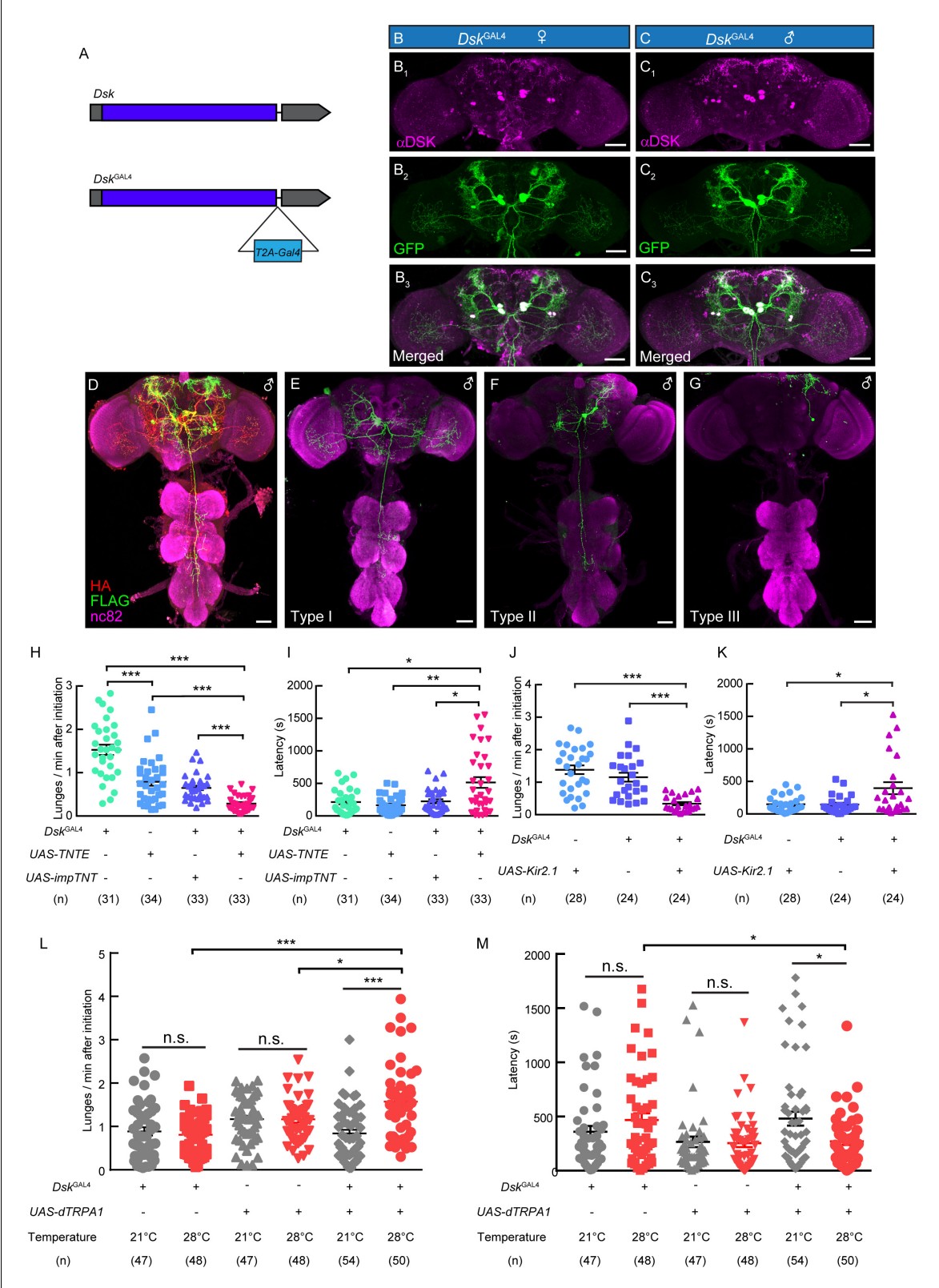

**Figure 2.** $Dsk^{GAL4}$ Neurons Modulate Male-Male Aggressive Behavior. (**A**) Generation of $Dsk^{GAL4}$ Knock-in line. GAL4 was fused to the end of open reading frame (ORF) of Dsk with a T2A peptide linker. (**B–C**) Anatomical features of DSK neurons revealed by $Dsk^{GAL4}$ driven *UAS-mCD8:GFP* expression. Adult female brains stained with anti-DSK antibody (magenta; **B₁**) and anti-GFP antibody (green; **B₂**); adult male brains stained with anti-DSK antibody (magenta; **C₁**) and anti-GFP antibody (green; **C₂**). Scale bars represent 50 μm. (**D–G**) Characterization of individual $Dsk^{GAL4}$ neurons using

*Figure 2 continued on next page*

*Figure 2 continued*

MultiColor FlpOut (MCFO) method in males. Eight DSK-expressing neurons were classified into three neuronal cell types (D–G). Scale bars represent 50 µm. (H–I) TNTE inactivation of $Dsk^{GAL4}$ neurons in males reduced lunge frequency (H) and prolonged fighting latency (I). (J–K) Kir2.1 inactivation of $Dsk^{GAL4}$ neurons in males decreased lunge frequency (J) and prolonged fighting latency (K). (L–M) Number of lunges per minute after initiation (L) and fighting latency (M) for males during thermogenetic activation of $Dsk^{GAL4}$ neurons. *p<0.05, **p<0.01, ***p<0.001, n.s. indicates no significant difference (Kruskal-Wallis and post-hoc Mann-Whitney U tests).

The online version of this article includes the following figure supplement(s) for figure 2:

**Figure supplement 1.** Projection patterns of $Dsk^{GAL4}$ neurons in female.

**Figure supplement 2.** Locomotion behavior of TNT and trpA1 experiments.

**Figure supplement 3.** The type I and type II neurons are necessary for aggression.

**Figure supplement 4.** The design of aggression chamber.

analyzing of type I or type II neurons separately also showed very similar results (*Figure 2—figure supplement 3B–C*). And these results should not be attributing to the inactivation of type III neurons, since no significant changes of lunge frequency have been observed after analyzing type III neurons. (*Figure 2—figure supplement 3D*). In addition, our electrophysiology experiments have shown that the activation of R71G01-LexA labeled P1 neurons (*Pan et al., 2012*) could strongly activates type I and type II neurons but only elicited weak responses from type III neurons. Those results demonstrated the necessity type I and type II neuron in aggression. But we still do not have enough evidences to show whether type III neurons are also involved in aggression.

## The *Dsk* receptor *CCKLR-17D1* regulates aggression

Two putative *Dsk* receptors have been identified in *Drosophila*: *CCKLR-17D1* and *CCKLR-17D3* (*Kubiak et al., 2002*; *Nässel and Williams, 2014*; *Nichols et al., 1988*). Loss-of-function mutants for *CCKLR-17D1* and *CCKLR-17D3* were generated by replacing the last two exons with 3P3-RFP cassette via homologous recombination (*Figure 3A*). Knock-in GAL4s for these two receptors were generated by fusing GAL4 to the C terminus of the receptors with a T2A linker to visualize the expression patterns of *Dsk* receptors (*Figure 3B*). We confirmed the targeted insertion with DNA sequencing. The knock-out lines were also validated with quantitative PCR (*Figure 3—figure supplement 1*). UAS-mCD8-GFP reporter driven by *CCKLR-17D1*$^{GAL4}$ marked neuronal clusters in the central complex, lateral horn, optical lobe, suboesophageal ganglion and ventral nerve cord (*Figure 3B*), while *CCKLR-17D3*$^{GAL4}$ predominantly labels neuronal clusters in the central complex, suboesophageal ganglion and ventral nerve cord (*Figure 3—figure supplement 2*).

Aggressive behavior was largely abolished in Δ*CCKLR-17D1* mutants compared with wildtype controls, while Δ*CCKLR-17D3* mutants preserved normal level of aggression (*Figure 3C and D*), and no significant locomotion defects were observed for Δ*CCKLR-17D1* and Δ*CCKLR-17D3* mutants (*Figure 3—figure supplement 3*). The reduced aggression phenotype could be rescued by pan-neuronal expression of *UAS-CCKLR-17D1* driven by *elav-GAL4* (*Figure 3E and F*). Moreover, RNAi silencing of *CCKLR-17D1* using *elav-GAL4* or *CCKLR-17D1*$^{GAL4}$ also significantly suppressed aggression (*Figure 3—figure supplement 4*). Taken together, these results suggest that Dsk act upon the CCKLR-17D1 receptor to modulate aggression.

To evaluate whether the aggression-promoting effect of activating $Dsk^{GAL4}$ neuron requires *CCKLR-17D1*, we used *UAS-dTRPA1* to activate $Dsk^{GAL4}$ neuron in the Δ*CCKLR-17D1* mutant background. In this case, dTRPA1 activation of $Dsk^{GAL4}$ neuron no longer enhanced aggression level in the absence of *CCKLR-17D1* (*Figure 3G and H*). We conclude that *CCKLR-17D1* is necessary for Dsk-mediated neuromodulation of aggressive behavior.

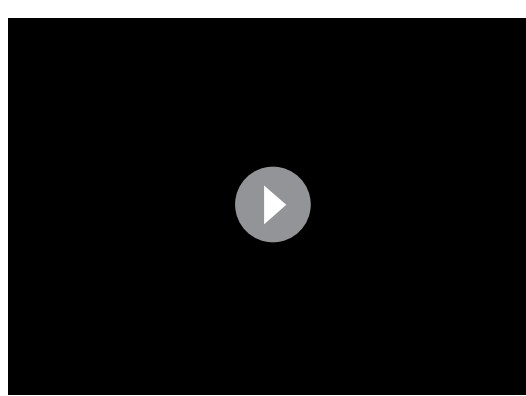

**Video 2.** Thermogenetic activation of $Dsk^{GAL4}$ neurons with UAS-dTRPA1 promotes inter-male aggression. https://elifesciences.org/articles/54229#video2

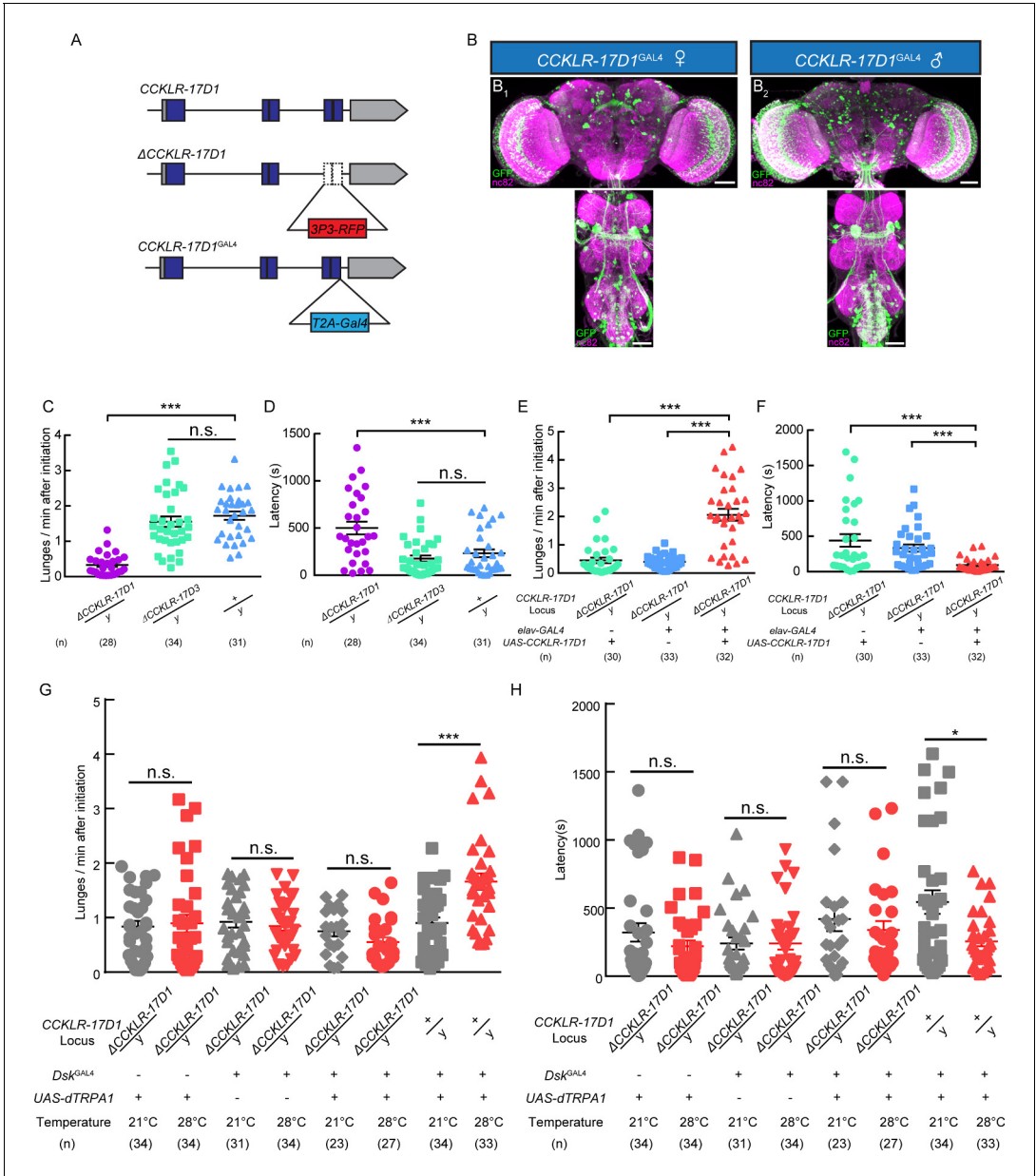

**Figure 3.** *Dsk* Receptor CCKLR-17D1 Is Necessary for Male-Male Aggression. (**A**) Schematic view of *CCKLR-17D1* gene locus and generation of *ΔCCKLR-17D1* and *CCKLR-17D1*[GAL4] Knock-in lines. The last two exons of *CCKLR-17D1* gene were replaced by *3P3-RFP* to generate *ΔCCKLR-17D1*. To generate *CCKLR-17D1*[GAL4] Knock-in, T2A-GAL4 was fused to the end of ORF. (**B**) Anatomical analysis of CCKLR-17D1 expressing neurons revealed by *CCKLR-17D1*[GAL4] driven *UAS-mCD8:GFP* expression. (**B₁**) Adult female CNS stained with anti-GFP antibody (green) and nc82 antibody (magenta). (**B₂**) Adult male CNS stained with anti-GFP antibody (green) and nc82 antibody (magenta). Scale bars represent 50 μm. (**C**) Number of lunges per minute after initiation of the indicated genotypes. (**D**) Fighting latency of the indicated genotypes. (**E and F**) Number of lunges per minute after initiation (**E**) and fighting latency (**F**) in *ΔCCKLR-17D1* mutants rescued by *elav-GAL4* driven *UAS-CCKLR-17D1* expression. (**G and H**) Number of lunges per minute after initiation (**G**) and fighting latency (**H**) during thermogenetic activation of *Dsk*[GAL4] neurons in the *ΔCCKLR-17D1* mutant background. Thermogenetic activation of *Dsk*[GAL4] neurons could not enhance aggression level in the *ΔCCKLR-17D1* mutant background. *p<0.05, **p<0.01, ***p<0.001, n.s. indicates no significant difference (Kruskal-Wallis and post-hoc Mann-Whitney U tests).

The online version of this article includes the following figure supplement(s) for figure 3:

**Figure supplement 1.** Molecular validation of mutants for *Dsk* receptors.

**Figure supplement 2.** Expression patterns of *CCKLR-17D3*[GAL4].

**Figure supplement 3.** *ΔCCKLR-17D1* does not affect locomotion behavior.

**Figure supplement 4.** Aggression, locomotion behavior of *CCKLR-17D1* RNAi in male flies.

**Figure supplement 5.** The design of aggression chamber.

## Dsk-expressing neurons function downstream of a subset of P1 neurons (*P1ᵃ-splitGAL4*) to Control Fighting Behavior

Previous works revealed that several group of neurons including a subset of P1 neurons (*Hoopfer et al., 2015*), Fru⁻ population of pC1 neurons (*Koganezawa et al., 2016*), pCd neurons (*Jung et al., 2020*), a small subset of aSP2 neurons (*Watanabe et al., 2017*), and a group of Fru⁺ Tk neurosn (*Asahina et al., 2014*) in the central brain can promote aggression when activated. Our anterograde trans-synaptic tracing experiments using *trans*-Tango method (*Talay et al., 2017*) revealed that there might be direct connections between a subset of P1 neurons and Dsk-expressing neurons. A split-GAL4 labeling a subset of P1 neurons was used to drive the expression of *trans*-Tango ligand and myrGFP, and the post-synaptic neurons were identified by immunostaining with anti-HA antibody (*Figure 4A*). Interestingly, a few post-synaptic neurons showed immunoreactivity to Dsk, suggesting that these Dsk-expressing neurons are post-synaptic to a subset of P1 neurons (*Figure 4A*, *Figure 4—figure supplement 1*). Furthermore, we confirmed the connection between a subset of P1 neurons and Dsk-expressing neurons using the GFP Reconstitution Across Synaptic Partners (GRASP) method (*Alekseyenko et al., 2014*; *Feinberg et al., 2008*). Significant GRASP signals were observed between R71G01-LexA labeled P1 neurons and DSK neurons (*Figure 4B*, *Figure 4—figure supplement 1*; *Pan et al., 2012*). Moreover, Anatomical registration of a subset of P1 neurons and DSK neurons suggests that a subset of P1 neurons axons overlap with DSK dendrites (*Figure 4C*).

Is Dsk necessary for the aggression promoting effects of a subset of P1 neurons? To address this question, a subset of P1 neurons were activated by *P1ᵃ split-GAL4* driving *UAS-dTRPA1* in the homozygous Δ*Dsk* mutant background. We observed that the aggression-promoting effect but not the courtship-promoting effect of a subset of P1 neurons activation is dramatically suppressed by the loss of the *Dsk* gene (*Figure 4F and G*, *Figure 4—figure supplement 2*). Taken together, these anatomical and behavioral analysis suggest that the Dsk system functions downstream of a subset of P1 neurons to modulate aggression.

## The functional connectivity between Dsk-expressing neurons and *R71G01-LexA* labeled P1 neurons

To investigate the functional connectivity between Dsk-expressing neurons and P1 neurons were labeled by *R71G01-LexA* (*Pan et al., 2012*), we examined whether and how Dsk-expressing neurons respond to the activation of P1 neurons by patch-clamp recordings and calcium imaging.

ATP is an extracellular messenger in organisms that can activate several cell-surface receptors, such as $P_2X_2$. The ATP-gated ionotropic purinoceptor $P_2X_2$ channel was expressed in P1 neurons for chemogenetic activation of P1 neurons by ATP (*Brake et al., 1994*). We performed patch-clamp recordings from DSK-L neurons (with cell bodies located laterally, type III) and DSK-M neurons (with cell bodies located in the medial region, type I and type II) (*Figure 5A*). In perforate patch recordings, ATP activation of P1 neurons elicited robust electrical responses from DSK-M neurons and relatively weak responses from DSK-L neurons (*Figure 5B*). Meanwhile, we also performed calcium imaging on DSK-M and DSK-L neurons expressing the calcium sensor GCaMP6m (*Figure 5C*). Consistently, ATP stimulation elicited a strong calcium response in DSK-M neurons and a smaller calcium response in DSK-L neurons (*Figure 5D* and *Video 3*). In contrast, ATP activation of DSK neurons did not induce calcium responses in P1 neurons (*Figure 5—figure supplement 1*). These data further confirmed the functional connections between Dsk-expressing neurons and P1 neurons, suggesting that Dsk-expressing neurons receive inputs from P1 neurons.

## Dsk signaling is upregulated in winner flies

It was well-known that hierarchical relationships could be established during fly fighting: the winners dominate the food patch and maintain high levels of aggressiveness, while the losers retreat and exhibit reduced aggression (*Yurkovic et al., 2006*). To explore the involvement of Dsk in the establishment of dominance, we used *elav-GS* to drive *UAS-Dsk* to induce conditional overexpression of Dsk. A pair of flies were introduced into one chamber, one of which overexpressed DSK induced by RU486 while the other did not. The one with Dsk overexpression tends to show increased aggression and establish social dominance over the opponent. It suggests that conditional overexpression of Dsk can promote social dominance (*Figure 6A–D*).

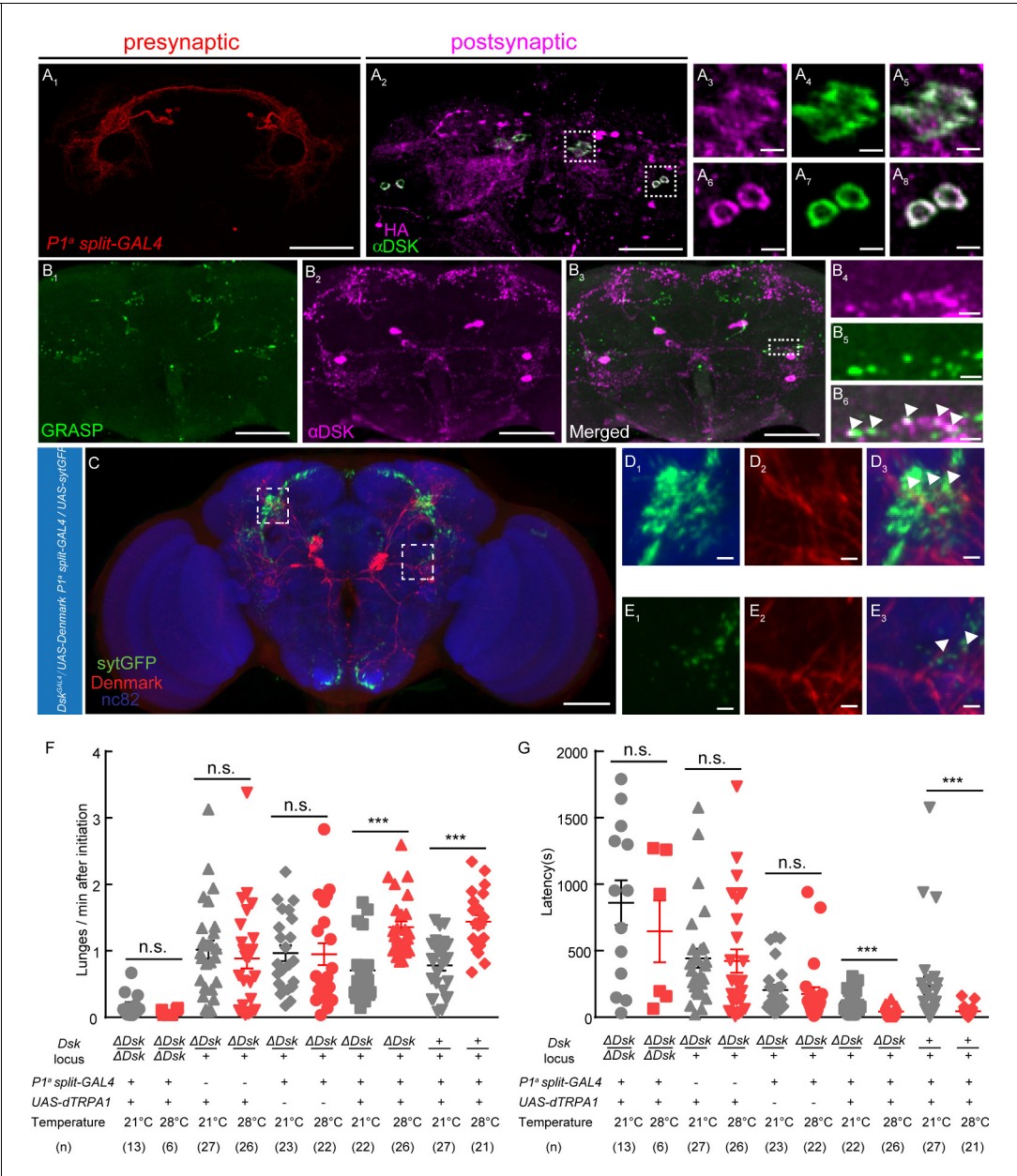

**Figure 4.** DSK Functions Downstream of a subset of P1 Neurons to Modulate Fighting Behavior. (A) Transsynaptic labeling by *trans*-Tango method identifies DSK neurons as postsynaptic partners of a subset of P1 neurons. Expression of the *trans*-Tango ligand in a subset of P1 neurons (red) (A₁) induces postsynaptic signals (anti-HA, magenta) (A₂) in the central brain. Green, anti-DSK antibody staining (A₂). (A₃–A₅) and (A₆–A₈) are enlargements of dashed boxes in (A₂). Genotypes of a subset of P1 neurons *(P1ᵃ split-GAL4): +/y; R15A01.AD/R15A01.AD; R71G01-GAL4.DBD/R71G01-GAL4.DBD*. (B) GRASP signals (B₁) reveal synaptic connections between *R71G01-LexA* labeled P1 Neurons and DSK neurons. spGFP1–10 is expressed by Dsk^GAL4 drivers; spGFP11 is driven by the *R71G01-LexA*. Magenta, anti-DSK antibody. (B₄–B₆) are enlargements of the dashed box in (B₃). White arrowheads point to areas in which GRASP signal co-localized with synaptic boutons revealed by anti-DSK antibody staining (B₆). Scale bars represent 50 μm in (A₁–A₂, B₁–B₃). Scale bars represent 5 μm (A₃–A₈, B₄–B₆). (C) Dendrites of DSK neurons revealed by *Dsk*^GAL4 driven *UAS-Denmark* expression (red). Axons of a subset of P1 neurons revealed by *P1ᵃ split-GAL4* driven *UAS-sytGFP* expression (green). nc82 antibody (blue) was used to label neuropils. Scale bars represent 50 μm. (D–E) Enlargements of dashed boxes in (C). White arrowheads point to areas in which a subset of P1 axons and DSK dendrites overlap. Scale bars represent 5 μm. (F and G) Number of lunges per minute after initiation (F) and fighting latency (G) during thermogenetic activation of a subset of *P1 neurons* in the Δ*Dsk* mutant background. The aggression-promoting effect of activating a subset of P1 neurons is suppressed by the Δ*Dsk* mutants. ***p<0.001, n.s. indicates no significant difference (Kruskal-Wallis and post-hoc Mann-Whitney U tests).

The online version of this article includes the following figure supplement(s) for figure 4:

**Figure supplement 1.** Controls for the trans-Tango and GRASP experiments.

**Figure supplement 2.** Activation of a subset of P1 neurons could promote courtship in the Δ*Dsk* mutant background.

*Figure 4 continued on next page*

*Figure 4 continued*

**Figure supplement 3.** The design of aggression chamber.

To assess the relationship between the activity of Dsk-expressing neurons and winner effect, we used the transcriptional reporter of intracellular Ca$^{2+}$ (TRIC) technique to monitor the neuronal activity (*Gao et al., 2015*). We found that the TRIC signals in the DSK-M neurons are significantly higher in the winner brains than in the loser and control brains (*Figure 6E–H*). Thus DSK-M neurons appear to be hyperactive in winners. It should be noted that TRIC signals are not detectable in the DSK-L neurons, possibly due to low levels of baseline activity in the DSK-L neurons.

## Discussion

In this study, we systematically dissected the neuromodulatory roles of the Dsk system in fly aggression. At the molecular level, Dsk neuropeptide and its receptor CCKLR-17D1 are important for fly aggression. At the circuit level, Dsk-expressing neurons function downstream of a subset of P1 neurons (*P1$^a$-splitGAL4*, 8 ~ 10 pairs of P1 Neurons) to control aggression. Furthermore, winners show increased calcium activity of Dsk-expressing neurons. Conditional overexpression of Dsk promotes winner effects, suggesting that Dsk is closely linked to the establishment of dominance. Taken together, our results elucidate the molecular and circuit mechanism underlying male aggression and suggest that cholecystokinin-like neuropeptide is likely to be evolutionarily conserved for the neuromodulation of aggression.

### Neural circuitry that controls male aggression

A neural circuitry controlling aggression should be composed of multiple modules that extend from sensory inputs to motor outputs. A variety of peptidergic and aminergic neurons are implicated in fly aggression (*Asahina, 2017*), but it is not clear how these modulatory neurons integrate input signals from other neural circuits to signal specific physiological states. Our data from circuit tracing, functional connectivity and behavioral epistasis suggest that Dsk-expressing neurons function downstream of a subset of P1 neurons and likely summate inputs from a subset of P1 neurons to signal an internal state of aggression (*Figures 4* and *5*, *Figure 5—figure supplement 1*). Activation of a subset of P1 neurons triggers both aggression and courtship (*Hoopfer et al., 2015*). Interestingly, while the aggression-promoting effect of activating a subset of P1 neurons is dramatically suppressed by the loss of the Dsk gene, the courtship-promoting effect remains intact in the ΔDsk mutant background. On the other hand, recent study suggested that Dsk neurons might function to antagonize P1 neurons on regulating male courtship (*Sf et al., 2019*). This dissociation suggests that while a subset of P1 neurons signal an arousal state facilitating both aggression and courtship, the Dsk system acts downstream of a subset of P1 neurons specifically required for aggression. It worth nothing to mention that the P1$^a$-splitGAL4 used in those studies not only labeled a small subset of Fru$^+$ neurons but also several Fru$^-$ neurons, and previous study on pC1 neurons suggested that Fru$^+$ pC1 neurons promote courtship and Fru$^-$ pC1 neurons promote aggression (*Koganezawa et al., 2016*), so further studies are needed to characterize whether different subset of P1$^a$-splitGAL4 labeled neurons are function differently on aggression and how Dsk system are involved. In addition, it remains unknown whether the Dsk system is responsible for integrating the sensory inputs and arousal state related to aggression, and how it connects to other components of the aggression circuitry, such as Tk neurons and AIP neurons.

As a caveat, it has been reported that Dsk is involved in feeding behavior (*Nässel and Williams, 2014*; *Williams et al., 2014*). Our experiment also reproduced the result that ΔDsk mutants show increased food consumption in the CAFE essay (*Diegelmann et al., 2017*). Previous studies reported a positive correlation between the body size of flies and the aggression level, suggesting that the modulational effects of DSK neurons on aggression and feeding can be separated (*Hoffmann, 1987*; *Hoffmann, 1990*). Further research is required to disentangle the relationship between DSK neurons modulating aggression and those regulating feeding.

In this study, we classified the eight DSK neurons into three subtypes (Type I, II and III) based on the morphology of the neurites or two subtypes (DSK-M and DSK-L) based on the location of the

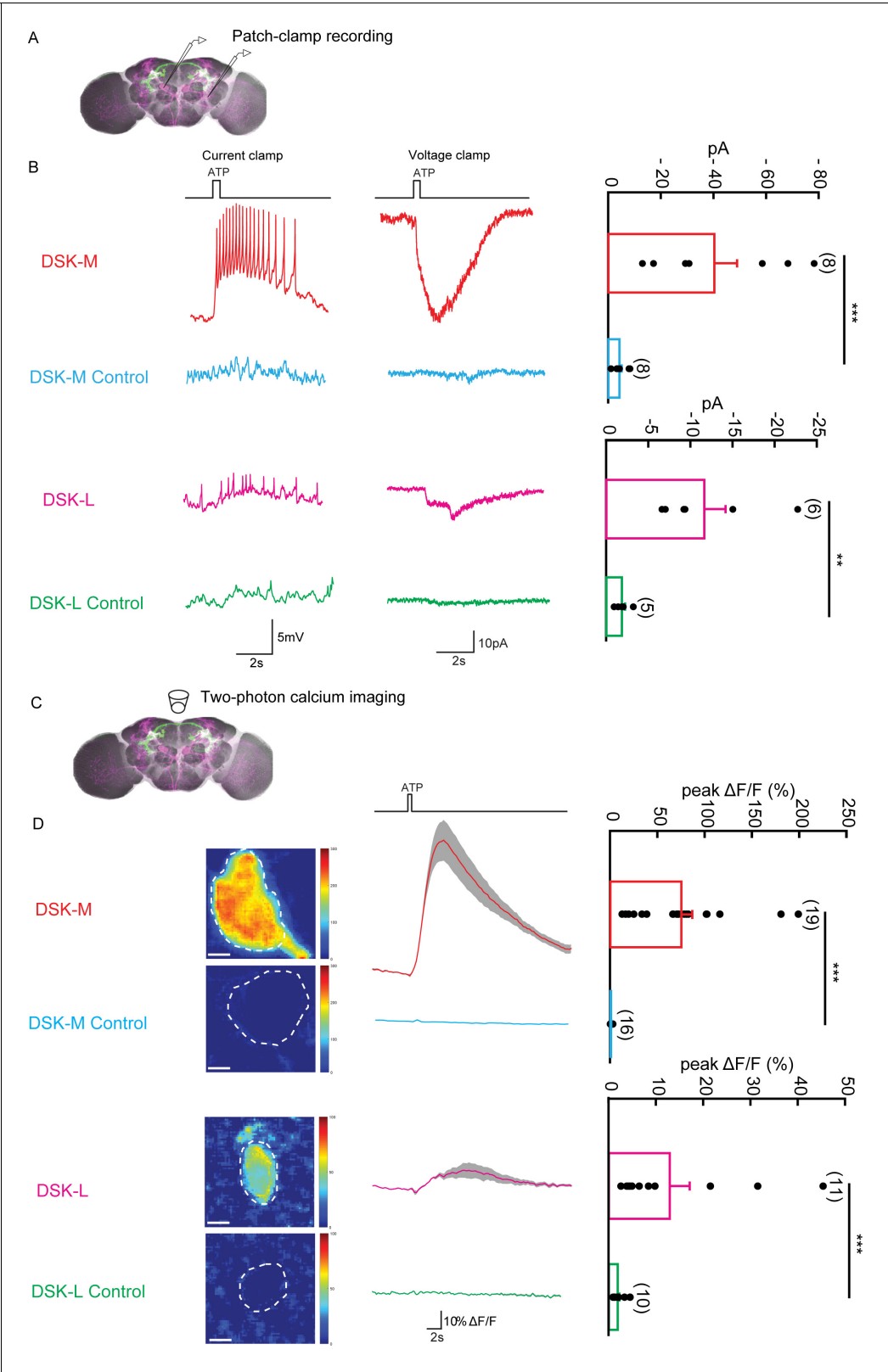

**Figure 5.** The Functional Connectivity Between DSK Neurons and *R71G01-LexA* labeled P1 Neurons. (**A**) Illustration of patch-clamp recording on DSK neurons (magenta). ATP-gated ion channel $P_2X_2$ was expressed in *R71G01-LexA* labeled P1 neurons for chemogenetic activation of P1 neurons. Genotype: *+/y; R71G01-LexA/+; Dsk$^{GAL4}$/LexAop-P2 × 2,UAS-GCaMP6m.* (**B**) The electrical responses of medial DSK neurons (DSK-M) and lateral DSK neurons (DSK-L) to ATP activation of *R71G01-LexA* labeled P1 neurons. ATP: 2.5 mM. Left: spikes firing (current clamp). Middle: current responses

*Figure 5 continued on next page*

*Figure 5 continued*

(voltage clamp). Right: quantification of absolute current responses. n = 8 from six flies for DSK-M, 8 from five flies for DSK-L, 6 from four flies for DSK-M control, 4 from four flies for DSK-L control. The electrical responses of medial DSK-M and DSK-L control to ATP activation. The Genotype of the control: *+/y; +/+; Dsk*$^{GAL4}$*/LexAop-P2 × 2,UAS-GCaMP6m.* (C) Illustration of two-photon calcium imaging in DSK neurons. (D) Left: Calcium imaging of GCaMP6m in the cell bodies of medial DSK neurons (DSK-M) and lateral DSK neurons (DSK-L) with P$_2$X$_2$-expressing *R71G01-LexA* labeled P1 neurons activated by ATP. ATP: 2.5 mM. Middle: calcium responses of DSK neurons to ATP stimulation. Gray envelopes represent SEM. Right: quantification of peak ΔF/F in the DSK-M and DSK-L neurons. Scale bars represent 5 μm. n = 19 from four flies for DSK-M, 16 from four flies for DSK-L, 11 from four flies for DSK-M control, 10 from four flies for DSK-L control. Genotypes: *+/y; R71G01-LexA/+; Dsk*$^{GAL4}$*/LexAop-P2 × 2, UAS-GCaMP6m* for DSK-M and DSK-L; *+/y; +/+; Dsk*$^{GAL4}$*/LexAop-P2 × 2, UAS-GCaMP6m* for DSK-M control and DSK-L control. ***p<0.001 (Mann-Whitney U tests).

The online version of this article includes the following figure supplement(s) for figure 5:

**Figure supplement 1.** *R71G01-LexA* labeled P1 neurons do not respond to activation of DSK neurons.

cell bodies. Interestingly, these subtypes also show functional difference in modulating aggression and differential connectivity with the a subset of P1 neurons. Note that Type I and II neurons correspond to DSK-M and Type III neurons correspond to DSK-L. Our finding that DSK-M neurons showed stronger responses to a subset of P1 neurons activation is consistent with the behavioral results of the flip-out experiment, in which Type I and II neurons, but not Type III, are critical to aggression. In future research, it would be interesting to use intersect method to more specifically label and manipulate the DSK neuron subtypes.

## Cholecystokinin-Like peptide is a potentially conserved mechanism underlying aggression

Previous study implicated that the cholecystokinin system is closely linked with various human psychiatric disorders, such as bipolar disorder (*Sears et al., 2013*) and panic attacks (*Bradwejn et al., 1990*). Interestingly, verbal aggression is promoted by the administration of cholecystokinin tetrapeptide in human subjects (*Tõru et al., 2010*). In cats, cholecystokinin agonists potentiate the defensive rage behavior while the cholecystokinin antagonists suppress it (*Luo et al., 1998*). Our results reveal that cholecystokinin-like peptide Dsk and Dsk receptor CCKLR-17D1 are important for *Drosophila* aggression (*Figures 1* and *3*). In addition, increased calcium activity in Dsk-expressing neurons coincides with winner states (*Figure 6*). Thus, the cholecystokinin system is linked to aggressive behavior in a variety of species and is likely to be an evolutionarily conserved pathway for modulating aggressiveness.

## Social hierarchy during fly fights

It has long been noticed that hierarchical relationships could be established during fly fights, with winners remaining highly aggressive and winning the subsequent encounters, and losers retreating and losing second fights (*Chen et al., 2002*; *Yurkovic et al., 2006*). The winner state is perceived as a reward signal while losing experience is aversive (*Kim et al., 2018*). The establishment of social hierarchy is only observed in males (*Nilsen et al., 2004*), and this male-specific feature of fly aggression is specified by fruitless (*Vrontou et al., 2006*). However, neural correlates of dominance have not been reported. In this study, Using a transcriptional reporter of intracellular calcium (TRIC), we found that winners display increased calcium activity in the median Dsk-expressing neurons. Moreover, conditional overexpression of Dsk specifically in the adult stage increases the flies' aggressiveness and makes them more likely to win against opponents without Dsk overexpression. Thus, both the enhanced Dsk signaling in the brain and the winning-promoting effect of conditional overexpression supported that the Dsk system may be involved in the establishment of social hierarchy during fly aggression.

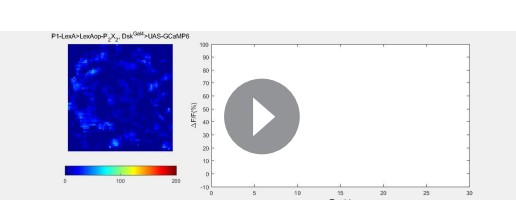

**Video 3.** Chemogenetic activation of P1 neurons by ATP elicits calcium responses from *Dsk*$^{GAL4}$ neurons.
https://elifesciences.org/articles/54229#video3

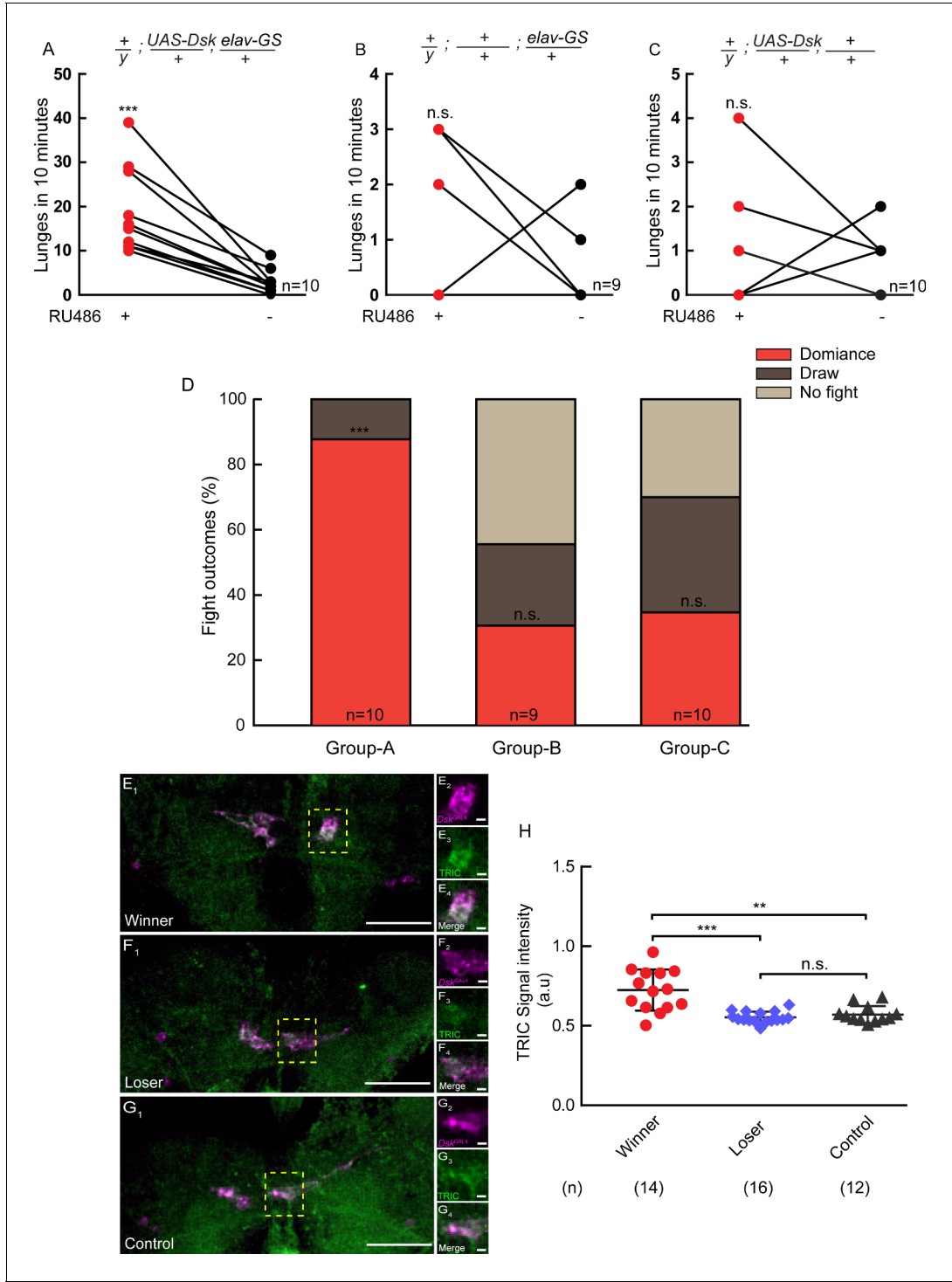

**Figure 6.** Conditional overexpression of DSK promotes winner effect and Winner Show Increased Calcium Activity of DSK-M Neurons. (**A–C**) Conditionally overexpression of Dsk increases number of lunges compared to controls. Pairs of flies of the indicated genotypes were introduced into one chamber, one of which overexpressed DSK induced by RU486 while the other did not. (**D**) Conditionally overexpression of Dsk promotes social dominance. Fight outcomes of the RU486-induced flies of the indicated genotypes in (**A–C**) were classified into 'dominance', 'draw' and 'no fight'. (**E–G**) Calcium activity of DSK-M neurons in winner (**E**), loser (**F**) and control (**G**) brains detected by the TRIC method. TRIC signal (green) in DSK-M neurons is increased in the winner brain (**E₃**). (**E₂–E₄**), (**F₂–F₄**), (**G₂–G₄**) are within the dashed boxes in (**E₁**), (**F₁**) and (**G₁**). Scale bars represent 50 μm. Genotype: *UAS-IVS-mCD8::RFP, LexAop2-mCD8::GFP/y; nSyb-MKII::nlsLexADBDo/+; Dsk^{GAL4}/UAS-p65AD::CaM*. (**H**) Quantification of TRIC signals in the DSK-M neurons of winner, loser and control brains. \*\*$p<0.01$, \*\*\*$p<0.001$, n.s. indicates no significant difference (Kruskal-Wallis and post-hoc Mann-Whitney U tests).

*Figure 6 continued on next page*

*Figure 6 continued*

The online version of this article includes the following figure supplement(s) for figure 6:

**Figure supplement 1.** The design of aggression chamber.

## Materials and methods

### Fly stocks

Flies were maintained on standard cornmeal medium at 25°C, 60% humidity in a 12 hr:12 hr Light: Dark cycle. For the thermogenetic activation experiments with *UAS-dTrpA1*, all the flies were reared at 21°C. The lines were backcrossed to isogenized Canton S flies for at least five generations prior to behavior studies. Trans-Tango lines were gifts from Dr. Yi Zhong (Tsinghua University). *+; sp/CyO; LexAop-P$_2$ × $_2$, UAS-GCamP/Tm2* was from Dr. Donggen Luo (Peking University). *UAS-TNTE* and *UAS-impTNT* were gifts from Dr. Cahir O'Kane (University of Cambridge). *UAS-dTRPA1* was a gift from Dr. Paul Garrity (Brandeis University). *R15A01-p65.AD* (BL#68837), *R71G01-GAL4.DBD* (BL#69507), *R71G01-LexA* (BL#54733), TRIC line (BL#61679), MCFO line (BL#64089), *UAS-Kir2.1* (BL#6595 and BL#6596), *UAS-mCD8::GFP* (BL#5137) and *LexAop2-mCD8::GFP* (BL#32203) were obtained from the Bloomington*Drosophila*Stock Center. *Lexo-CD4-spGFP11/CyO; UAS-CD4-spGFP1-10/Tb* was previously described (*Gordon and Scott, 2009*). Please refer to *Supplementary file 1* for the complete genotypes of fly stocks used in this study.

### Method details

#### Calcium imaging and electrophysiological recordings

Young adult flies (1–2 days after eclosion) were used for calcium imaging. Fly brain was dissected out with fine forceps in saline solution. The saline is composed of (in mM): 103 NaCl, 3 KCl, 4 MgCl$_2$, 1.5 CaCl$_2$, 26 NaHCO$_3$, 1 NaH$_2$PO$_4$, 5 N-tri-(hydroxymethyl)-methyl-2-aminoethane-sulfonic acid (TES), 20 D-glucose, 17 sucrose, and five trehalose. The brain was continuously perfused with saline saturated with 95% O$_2$/5% CO$_2$ (~pH 7.3) at 21°C during imaging.

Time series of images were acquired on a Nikon A1R+ confocal microscope with a 60 × water immersion objective. A 512 × 128 pixel imaging region was captured at a frame rate of 7 fps. The images were analyzed with a graphical user interface (GUI) written in Matlab (*Zhou et al., 2015*). Regions of interest (ROIs) were selected manually, and the mean baseline fluorescence values (F$_0$) were calculated from 30 frames before ATP stimulation. The fluorescence change was calculated as: $\Delta F/F = (F_t - F_0)/F_0 \times 100\%$. Peak $\Delta F/F$ is defined as the maximum $\Delta F/F$ value after the onset of ATP stimulus.

Electrophysiological recordings were conducted on the same microscope with infrared-differential interference contrast (IR-DIC) optics and IR-CCD (DAGE-MTI) for visualizing the target neurons. The recording pipette (~10–15 MΩ) was filled with internal solution containing 150 μg/ml amphotericin B. The internal solution consists of (in mM): 140 K-gluconate, 6 NaCl, 2 MgCl$_2$, 0.1 CaCl$_2$, 1 EGTA, 10 HEPES (pH 7.3). Signals were amplified with MultiClamp 700B, digitized with Digidata 1440A, recorded with Clampex 10.6 (all from Molecular Devices), filtered at 2 kHz, and sampled at 5 kHz. The recorded neuron was voltage clamped at −70 mV. Measured voltages were corrected for a liquid junction potential.

### Chemogenetic stimulation

ATP-gated ion channel P$_2$X$_2$ was expressed in P1 neurons. 2.5 mM ATP-Na (Sigma-Aldrich) was delivered through a three-barrel tube, controlled by SF77B stepper (Warner Instruments) driven by Axon Digidata 1440A analog voltage output. These devices achieved fast switch between perfusion saline and ATP stimulation.

### Immunohistochemistry

The adult flies were cold anesthetized and the CNSs were dissected in 1X PBS (Corning, 21–040-CVR). The samples were fixed in 2% paraformaldehyde for 55 min at room temperature, washed three times for 20 min in PBT (1X PBS containing 0.3% Triton-X100) at room temperature, blocked in PBT containing 5% goat serum for 1 hr at room temperature. Brains were then incubated with

primary antibody (diluted in blocking solution) for 18 ~ 24 hr at 4°C, washed three times in 0.3% PBT for 20 min before incubated in the secondary antibody (diluted with 5% goat serum in 0.3%PBT) for 18 ~ 24 hr at 4°C. Brain samples were then washed three times for 20 min in PBT at room temperature, fixed in 4% paraformaldehyde for 4 hr at room temperature, and washed in PBT three times for 20 min at room temperature. Lastly, brains were mounted onto poly-L-lysine (PLL)-coated coverslip in 1X PBS. The coverslip with mounted brains was then soaked for 5 min each in a gradient of ethanol baths: 30%, 50%, 75%, 95%, 100%, and then soaked three times for 5 min in xylene. DPX was applied to the samples on the coverslip and the coverslip was placed on the slide and dried for 2 days before imaging. Images were taken with Carl Zeiss (LSM710) confocal microscopy and then processed with Fiji software. The following primary antibodies were used: mouse anti-nc82 (1:50; DSHB), chicken anti-GFP (1:1000; life technologies), rabbit anti-DSK antibody (1:1000), rabbit anti-GFP (1:1000; Invitrogen), rabbit anti-HA (1:300; Cell Signaling Technologies), rat anti-FLAG (1:200; Novus Biologicals), Rat anti-HA (1:100; Roche), mouse anti-GFP-20 (1:100; sigma), rabbit anti-RFP (1:1000; Invitrogen). The following secondary antibodies were used: Alexa Fluor goat anti-chicken 488 (1:500; life technologies), Alexa Fluor goat anti-rabbit 488 (1:500; life technologies), Alexa Fluor goat anti-mouse 546 (1:500; life technologies), Alexa Fluor goat anti-rat 546 (1:500; Invitrogen), Alexa Fluor goat anti-rabbit 546 (1:500; Invitrogen), Alexa Fluor goat anti-rat 633 (1:500; Invitrogen), and Alexa Fluor goat anti-rat 647 (1:500; Invitrogen), (1:500; life technologies).

## Brain registration

CMTK software was used to generate a standard brain by averaging six male brains and six female brains stained with nc82 antibody (*Rohlfing and Maurer, 2003*). Confocal stacks were then registered into the common standard brain by linear registration and non-rigid warping methods based on the nc82 channel (*Jefferis et al., 2007*).

## Generation of anti-DSK antibody

The antisera used to recognize Dsk peptide were raised in New Zealand white rabbits using the synthetic peptide N'-GGDDQFDDYGHMRFG-C'. Because Dsk and FMRFamide peptides share the same C-terminal sequence, antibody against Dsk may also recognize FMRFamide peptides. To resolve this problem, we negatively purified the antiserum by binding the antiserum to a FMRFamide peptide affinity column after positive purification. The synthesis of antigen peptide and FMRF peptide, the production and purification of antiserum were performed by Beijing Genomics Institute (BGI).

## Generation of Knock-Out and Knock-In lines

We generated mutants of Dsk, CCKLR17D1 and CCKLR17D3 through homologous recombination in *Drosophila* embryos with the CRISPR/Cas9 system.

When generating the knock-out mutants, we introduced an attP site into the noncoding region of each gene. Based on the knock-out lines, knock-in Gal4s for each gene were generated through phiC31 mediated transgenesis as described in *Deng et al. (2019)*.

To construct targeting vectors for knock-out generation, the homologous arms were subcloned into pBSK-attP-3P3-RFP-loxP vector through Gibson assembly using the following primers:

## Dsk (CG18090)

5' arm-F ctatagggcgaattgggtacTAGCTTGAGCTGCGTTTATG
5' arm-R cgccaactcgtagtatgcggccgcTGTATATGGGCATGAGGTTG
3'arm-F ccgcatactacgagttggcgcgccCGATAAACACTTGCCATCAG
3' arm-R aaaagctggagctccaccgcAGTTCGTTAGAGCAACGCCT

## CCKLR17D1 (CG42301)

5' arm-F ctatagggcgaattgggtacCCAACCGCAAACGGCAATAA
5' arm-R cgccaactcgtagtatgcggccgcGTCTCGAATCTTGCGTGATT
3' arm-F ccgcatactacgagttggcgcgccGTCACTTTAGGTTAGCAATG
3' arm-R aaaagctggagctccaccgcgagaagggagcgtcgtagtc

## CCKLR17D3 (CG32540)

5' arm-F ctatagggcgaattgggtacCAAGTTCCTCGAAGAGCGAC
5' arm-R cgccaactcgtagtatgcggccgcTGTACCCACACCCTGCCCAT
3' arm-F ccgcatactacgagttggcgcgccCTCTAAGCTGTAGAGGATTC
3' arm-R aaaagctggagctccaccgcCGAGTAGTTTGTCCTGTCAT

To generate knock-in Gal4 constructs, the deleted regions of each gene's knock-out mutant were subcloned into the pBSK-attB-2xMyc-T2A-Gal4 (*Deng et al., 2019*) through Gibson assembly using the following primers.

## Dsk

F: cccgggcgcgtactccacgcATGGTTCACAGCTCAGTTTA
R: ccacctccaccacccgcggccgcTCGGCCGAAACGCATGTGAC

## CCKLR17D1

F: cccgggcgcgtactccacgcAGCTAGCATTGGGCTTTGAA
R: ccacctccaccacccgcggccgcGAGTCGCGGACTCTCGAGGA

## CCKLR17D3

F: cccgggcgcgtactccacgcAATTGCTTTACAATGGGAAA
R: ccacctccaccacccgcggccgcGAGCTGAGGACTGTTGACGG

To generate $\Delta Dsk^{GAL4}$, the coding region of *Dsk* was deleted and replaced by *GAL4* cassette. First, the 551 bp before the coding region and coding region of *Dsk* was replaced by an attP-3P3-RFP cassette through homologous recombination to obtain the $\Delta Dsk$ knock-out line. $\Delta Dsk^{GAL4}$ flies were then generated through phiC31 mediated attB/attP recombination in the knock-out line such that the deleted 551 bp of non-coding sequence followed by GAL4 cassette was integrated into the *Dsk* locus. Primers for amplifying the 551 bp before the coding region of *Dsk*:

## $\Delta Dsk^{GAL4}$-F

GTACGCTAGCATGGTTCACAGCTCAGTTTAAC

## $\Delta Dsk^{GAL4}$-R

GTACGCGGCCGCACAGCTTCTAGGTCCCATG

The embryo injection and fly screen processes were the same as described in *Deng et al. (2019)*. We verified all the knock-out and knock-in lines through PCR amplification followed by DNA sequencing.

### Generation of *UAS-Dsk* and *UAS-CCKLR-17D1*

The pJFRC28-*UAS-Dsk* ('*UAS-Dsk*') and pJFRC28-*UAS-CCKLR-17D1* ('*UAS-CCKLR-17D1*') constructs were generated using pJFRC28-10XUAS-IVS-GFP-p10 (Addgene # 36431). The plasmid of pJFRC28-10XUAS-IVS-GFP-p10 was digested with NotI and XbaI to remove the GFP coding sequence, and then the cDNA of *Dsk* and *CCKLR-17D1* were cloned into this plasmid by Gibson Assembly. The Kozak sequence was added right upstream of the ATG. Next, *UAS-Dsk* and *UAS-CCKLR-17D1* constructs were injected and integrated into the attP40 and attP2 sites on the second and third chromosome through phiC31 integrase mediated transgenesis, respectively. All the transgenes were verified by PCR and DNA sequencing. The primer sets for cloning the cDNA of *Dsk* and *CCKLR-17D1*:

## UAS-Dsk-F

TCTTATCCTTTACTTCAGGCGGCCGCCACCATGGGACCTAGAAGCTGTACGCA

## UAS-Dsk-R

GTTATTTTAAAAACGATTCATTCTAGATTATCGGCCGAAACGCATGTGA

## UAS-CCKLR-17D1-F

TCTTATCCTTTACTTCAGGCGGCCGCCACCATGTTGCCGCGCCTGTGCGCCGACGCTT

## UAS-CCKLR-17D1-R

GTTATTTTAAAAACGATTCATTCTAGATCAGAGTCGCGGACTCTCGAGGATCGTGT

## Quantitative PCR

Whole head RNA was extracted from 50 fly heads using TRIzol (Ambion #15596018). The cDNA of whole fly heads was synthesized using Prime Script reagent kit (Takara).

Quantitative PCR was performed on Thermo Piko Real 96 (Thermo) using SYBR Green PCR Master Mix (Takara #RR820A). The primer sets used in this study are:

*Dsk*: 5'- GAACGCTAAGGATGATCGGC −3' and 5'- ATTACGCCTGTCCCCGAATAG −3'
*CCKLR-17D1*: 5'- TGAGCGACAATGAATCCC −3' and 5'- CTTGACCACACGCTTCTTG −3
*CCKLR-17D3*: 5'- GCCCATAGCGGTCTTTAGTC-3' and 5'- GTGATGAGGATGTAGGCCAC −3
*Actin 5C-PB*: 5'- CCAACCGTGAGAAGATGACC −3' and 5'- GCCGGAGTCCAGAACGATAC −3'

## Behavior assay

### Aggressive behavior

The flies were raised at 25℃ and 60% humidity under a 12 hr:12 h L:D cycle. Male or female flies were collected immediately after eclosion and isolated for 5–7 days in food tubes prior to behavioral assays. All behavioral assays were carried out at 25℃ and 60% humidity between 11AM-16PM, except the thermogenetic experiments.

In most experiments, aggression assays were done in the aggression chamber (*Figure 1—figure supplement 6*, *Figure 2—figure supplement 4*, *Figure 3—figure supplement 5*, *Figure 4—figure supplement 3*, *Figure 6—figure supplement 1*). The aggression chamber is made up of four acrylic plates. The bottom plate has 12 wells for containing food substrates (diameter: 8 mm; depth: 3 mm). The lower and upper plates have 12 cylindrical arenas (diameter: 15 mm; height of each plates: 3 mm). Two males were briefly cold anesthetized and introduced into the lower and upper plates separated by a transparent film. The flies were allowed to recover for 60 min at 25℃. Once the recording began, the transparent film was removed to allow aggressive encounters between the two flies. The aggressive encounters were videotaped for 30 min by cameras (Canon VIXIA HF R500) at 30 fps for further analysis. The number of lunges within 30 min and the latency of initiating fighting after removing the separation are counted. All behavioral assays were analyzed manually and assigned to three experimenters randomly for independent scoring. The scorers were blind to the genotypes and conditions of the experiment.

In dTRPA1 experiments, flies were reared at 21℃ and isolated for 6–7 days. Aggressive assay was carried out at 21℃ (control) or 28℃ (activation) and 60% humidity. For the activation group, the aggression chambers containing flies will be pre-warmed at 28℃ for 20 min prior to removing the separation and videotaping.

To generate mosaic flies and enable stochastic inactivation of DSK neurons, the flies of hs-FLP; UAS > stop > Kir2.1$^{eGFP}$/Dsk$^{GAL4}$ were heat-shocked for 90 min at 37℃ during the mid- to late-larval stage. Male flies were collected immediately after eclosion, marked on the thorax with yellow acrylic paint on the second day of eclosion, and isolated for 5–7 days in food tubes prior to behavioral assays. Each male mosaic fly was paired with a ΔDsk mutant which served as a hypo-aggressive

opponent. The homozygous $\Delta Dsk$ male mutants showed a significantly reduced lunge frequency. The mosaic flies could attack homozygous $\Delta Dsk$ mutants, and then we could obtain larger aggression scores from mosaic flies, so we choose $\Delta Dsk$ mutants as an attack target for mosaic flies. After testing individual male mosaics for aggression, we dissected and stained their brains using anti-GFP to identify the cell types labeled.

## Courtship behavior

The virgin of Females and males were anesthetized on ice and introduced into the lower and upper courtship chamber separated by a transparent film. The courtship chamber is cylindrical and consists of two layers (diameter: 10 mm; height of each layer: 3 mm). Flies were allowed to recover for 1 hr at 25℃. After 1 hr, the transparent film was removed and courtship video was videotaped for 30 min by cameras (Canon VIXIA HF R500) at 30 fps for further analysis.

## Experiments of social hierarchy

The virgin male flies were collected and isolated upon eclosion. On the second day of eclosion, flies were anesthetized under light $CO_2$ and marked on the thorax with yellow and red acrylic paint, respectively, and then singly housed in food tubes. On the sixth day after eclosion, a pair of flies with two different colors were introduced into one aggressive chamber and a transparent film was used to separate the two males. Meanwhile, we introduced a single painted fly into another chamber as the control group without fighting experience. Flies were allowed to recover for 60 min at 25℃ before removing the separation to allow aggressive encounters for a period of 4 hr (eight 30 min sessions). Usually at the end of the first session social dominances were stably established with winners occupying the food patch and lunging toward losers. We chose pairs of winners/losers that maintain stable dominance relationships in all the eight sessions for further TRIC analysis.

In TRIC experiments, the brains of these three groups were dissected and fixed with 8% paraformaldehyde for 2 hr, and then mounted with DPX. All the brains were imaged on Carl Zeiss (LSM710) confocal microscopy with the same settings.

We used Fiji to quantify the fluorescence intensity by first generating a Z stack of the sum of fluorescence signals, and then quantifying the total intensity of manually selected ROIs. The signal intensity was normalized to that of the control group.

## Statistical analysis

All statistical analyses were conducted using the Prism7 (GraphPad software) or MATLAB (MathWorks) software. Kruskal-Wallis ANOVA test followed by post-hoc Mann-Whitney U test, was used to identify significant differences between multiple groups. The Mann-Whitney U test was used for analyzing the data of two columns. The sample sizes are indicated in the figures.

## Acknowledgements

We thank Yi Zhong (Tsinghua University), Cahir O'Kane (University of Cambridge), Paul Garrity (Brandeis University) for sharing fly strains; Chenzhu Wang (Chinese Academy of Sciences), Yufeng Pan (Southeast University) and Jue Xie (Chinese Institute for Brain Research, Beijing) for their comments; Jing Li and Suyue Zhou for assistance with behavior assays; Yihui Chen for the maintenance of fly stocks; other members of the Zhou laboratory for helpful discussions. This work is supported by the Strategic Priority Research Program of the Chinese Academy of Science (No. XDB11010800), National Natural Science Foundation of China (No. 31872280, 31622054, 31671096), State Key Laboratory of Integrated Management of Pest Insects and Rodents, IOZ, CAS (No. Y652751E03, Y752781A03).

## Additional information

### Funding

| Funder | Grant reference number | Author |
| --- | --- | --- |
| National Natural Science Foundation of China | 31872280 | Chuan Zhou |

| | | |
|---|---|---|
| National Natural Science Foundation of China | 31622054 | Chuan Zhou |
| National Natural Science Foundation of China | 31671096 | Chuan Zhou |
| Chinese Academy of Sciences | XDB11010800 | Chuan Zhou |
| State Key Laboratory of Integrated Management of Pest Insects and Rodents | Y652751E03 | Chuan Zhou |
| State Key Laboratory of Integrated Management of Pest Insects and Rodents | Y752781A03 | Chuan Zhou |

The funders had role in study design, data collection and interpretation, and the decision to submit the work for publication.

## Author contributions

Fengming Wu, Resources, Data curation, Writing - original draft; Bowen Deng, Yining Li, Methodology; Na Xiao, Kai Shi, Software; Tao Wang, Resources; Rencong Wang, Data curation; Dong-Gen Luo, Yi Rao, Supervision; Chuan Zhou, Conceptualization, Funding acquisition, Writing - review and editing

## Author ORCIDs

Na Xiao https://orcid.org/0000-0002-7732-9117
Yi Rao https://orcid.org/0000-0002-0405-5426
Chuan Zhou https://orcid.org/0000-0001-7952-7048

## Decision letter and Author response

Decision letter https://doi.org/10.7554/eLife.54229.sa1
Author response https://doi.org/10.7554/eLife.54229.sa2

# Additional files

## Supplementary files

- Supplementary file 1. Key resources table.
- Supplementary file 2. The number of labeled cells of mosaic flies.
- Transparent reporting form

## Data availability

All data generated or analysed during this study are included in the manuscript and supplementary files. Source data files have been provided for main figures and supplementary figures.

The following previously published dataset was used:

| Author(s) | Year | Dataset title | Dataset URL | Database and Identifier |
|---|---|---|---|---|
| Nichols R, Schneuwly SA, Dixon JE | 1988 | Identification and Characterization of a Drosophila Homologue to the Vertebrate Neuropeptide Cholecystokinin | http://flybase.org/reports/FBgn0000500 | flybase, FBgn0000500 |

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
