## [Decision Letter]

**Acceptance summary:**

The authors present an extensive set of experiments to show that Drosulfakinin neurons are involved in *Drosophila* aggression, and suggest that this peptide family (which includes the mammalian ortholog Cholecystokinin) may have conserved behavioral functions. Specifically, the authors suggest that the Dsk system plays an important role in promoting aggression and the establishment of winner's state, acting downstream of P1 interneurons that have been implicated to be a trigger center for aggression and courtship.

**Decision letter after peer review:**

Thank you for submitting your article "A neuropeptide regulates fighting behavior in *Drosophila melanogaster*" for consideration by *eLife*. Your article has been reviewed by two peer reviewers, and the evaluation has been overseen by Leslie Griffith as Reviewing Editor and Ronald Calabrese as the Senior Editor. The reviewers have opted to remain anonymous. The reviewers have discussed the reviews with one another and the Reviewing Editor has drafted this decision to help you prepare a revised submission.

Summary:

The authors present an extensive set of experiments to show that Drosulfakinin neurons are involved in *Drosophila* aggression, and suggest that this peptide family (which includes the mammalian ortholog Cholecystokinin) may have conserved behavioral functions. Specifically, the authors suggest that the Dsk system plays an important role in promoting aggression and the establishment of winner's state, acting downstream of P1 interneurons that have been implicated to be a trigger center for aggression and courtship. In general, experiments were conducted carefully and the quality of data was high. There remain a few issues with data analysis and presentation, however, that require attention.

Essential revisions:

1) The analysis of lunging data may be obscuring the actual cause of the phenotype. Because of the way the data are presented, it is impossible to know if the low values they see are due to low levels of lunging or failure to initiate lunging within the 30 min window. The authors either need to reanalyze their data to look at lunge/min after initiation or collect a new data set that looks at the lunge rate AFTER initiation. This is an important point and the authors need to differentiate between initiation and maintenance as the basis of the defect.

2) The authors fail to provide any locomotor controls. This is critical since the authors are claiming the phenotypes are specific to aggression.

3) Some of the ephys/imaging data cannot be correct as shown. The upper traces in Figure 5B seem to indicate the time of light stimulation to release caged ATP, but the onset of ATP release as indicated in the figure preceded the initial sign of deflection of electrical responses, in particular the current response (lower trace). This invites the suspicion that the ATP trace was not correctly recorded along with the voltage/current trances. The time required for synaptic transmission (synaptic delay) is typically just a few milliseconds. To demonstrate direct synaptic connections, the postsynaptic response must be recorded together with the spike invading the presynaptic terminal. The presented data do not satisfy this requirement. The statement "These data further confirmed the synaptic connections between Dsk-neurons" should be toned down and the figure should be clarified/corrected.

4) In general there are problems with data presentation (see #1 above). The authors should present all the individual points in addition to the mean and confidence intervals. Their latency data might look quite bimodal if presented this way and that would be informative. It is also *eLife*'s policy that all the data should be available to the reader and this could easily be done here.

---

## [Author Response]

Essential revisions:1) The analysis of lunging data may be obscuring the actual cause of the phenotype. Because of the way the data are presented, it is impossible to know if the low values they see are due to low levels of lunging or failure to initiate lunging within the 30 min window. The authors either need to reanalyze their data to look at lunge/min after initiation or collect a new data set that looks at the lunge rate AFTER initiation. This is an important point and the authors need to differentiate between initiation and maintenance as the basis of the defect.

As the reviewers suggested, we have re-analyzed the data and redrawn all the figures and presented the behavioral data in the format of scatter plots.

2) The authors fail to provide any locomotor controls. This is critical since the authors are claiming the phenotypes are specific to aggression.

We analyzed the general locomotion activity of ΔDsk,ΔCCKLR-17D1 mutants and added to the manuscript as Figure 1—figure supplement 5 and Figure 3—figure supplement 3. We didn’t observed significant changes of locomotion activity for ΔDsk and ΔCCKLR-17D1 mutants.

We also analyzed the general locomotion activity of TNT and trpA1 experiments and add to the manuscript as Figure 2—figure supplement 2. Compared to control males, locomotion was not significantly different in males carrying (Dsk^GAL4^ and *UAS-TNT*) or carrying (Dsk^GAL4^ and UAS- trpA1).

3) Some of the ephys/imaging data cannot be correct as shown. The upper traces in Figure 5B seem to indicate the time of light stimulation to release caged ATP, but the onset of ATP release as indicated in the figure preceded the initial sign of deflection of electrical responses, in particular the current response (lower trace). This invites the suspicion that the ATP trace was not correctly recorded along with the voltage/current trances. The time required for synaptic transmission (synaptic delay) is typically just a few milliseconds. To demonstrate direct synaptic connections, the postsynaptic response must be recorded together with the spike invading the presynaptic terminal. The presented data do not satisfy this requirement. The statement "These data further confirmed the synaptic connections between Dsk-neurons" should be toned down and the figure should be clarified/corrected.

We thank the reviewer for pointing this out. We used perfusion system to carry out the experiment. Young adult flies (1-2 days after eclosion) were used for calcium imaging. Fly brain was dissected out with fine forceps in saline solution ATP-Na (Sigma-Aldrich) of 2.5mM was delivered through a three-barrel tube, controlled by SF77B stepper (Warner Instruments) driven by AxonTM Digidata 1440A analog voltage output. These devices achieved fast switch between perfusion saline and ATP stimulation. ATP-Na (Sigma-Aldrich) of 2.5mM was delivered through a three-barrel tube and it may take a certain amount of time to work, so there's a delay in the current and voltage response. We have modified the writing in the manuscript: "These data further confirmed the functional connections between Dsk-expressing neurons and P1 neurons".

4) In general there are problems with data presentation (see #1 above). The authors should present all the individual points in addition to the mean and confidence intervals. Their latency data might look quite bimodal if presented this way and that would be informative. It is also eLife's policy that all the data should be available to the reader and this could easily be done here.

We have redrawn all figures and presented the behavioral data in the format of scatter plots.